# Electrochemical Behavior of Symmetric Electrical Double-Layer Capacitors and Pseudocapacitors and Identification of Transport Anomalies in the Interconnected Ionic and Electronic Phases Using the Impedance Technique

**DOI:** 10.3390/nano12040676

**Published:** 2022-02-18

**Authors:** Willian G. Nunes, Aline M. Pascon, Bruno Freitas, Lindomar G. De Sousa, Débora V. Franco, Hudson Zanin, Leonardo M. Da Silva

**Affiliations:** 1Carbon Sci-Tech Labs, Center for Innovation on New Energies, Advanced Energy Storage Division, School of Electrical and Computer Engineering, University of Campinas, Av. Albert Einstein 400, Campinas 13083-852, SP, Brazil; nuneswillian40@gmail.com (W.G.N.); alinepascon@gmail.com (A.M.P.); brungaf@gmail.com (B.F.); 2Laboratory of Fundamental and Applied Electrochemistry, Department of Chemistry, Federal University of Jequitinhonha e Mucuri’s Valley, Rodovia MGT 367, Km 583, n° 5000, Alto da Jacuba, Diamantina 39100-000, MG, Brazil; lindogomes01@hotmail.com (L.G.D.S.); dvfranco@gmail.com (D.V.F.)

**Keywords:** impedance models, disordered electrode materials, anomalous charge transport, supercapacitors

## Abstract

A double-channel transmission line impedance model was applied to the study of supercapacitors to investigate the charge transport characteristics in the ionic and electronic conductors forming the electrode/solution interface. The macro homogeneous description of two closely mixed phases (Paasch–Micka–Gersdorf model) was applied to study the influence of disordered materials on the charge transport anomalies during the interfacial charge–discharge process. Different ex situ techniques were used to characterize the electrode materials used in electrical double-layer (EDLC) and pseudocapacitor (PC) devices. Two time constants in the impedance model were adequate to represent the charge transport in the different phases. The interfacial impedance considering frequency dispersion and blocked charge transfer conditions adequately described the charge storage at the interface. Deviations from the normal (Fickian) transport involving the ionic and electronic charge carriers were identified by the dispersive parameters (e.g., *n* and *s* exponents) used in the impedance model. The ionic and electronic transports were affected when the carbon-based electrical double-layer capacitor was converted into a composite with strong pseudocapacitive characteristics after the decoration process using NiO. The overall capacitance increased from 2.62 F g^−1^ to 536 F g^−1^ after the decoration. For the first time, the charge transport anomalies were unequivocally identified in porous materials used in supercapacitors with the impedance technique.

## 1. Introduction

According to the literature [1,2,3,4,5,6,7], the electrochemical properties exhibited by different types of porous/nanostructured carbon-based materials used in supercapacitors can be considerably improved after their decoration using different transition metal oxides (TMOs) (e.g., NiO, Co_3_O_4_, Nb_2_O_5_, NiCo_2_O_4_, and MnO_2_). However, as recently discussed by some of the present authors [1,8,9,10,11,12], the electrochemical characterization of composite materials containing different carbon structures and TMOs is not easy due to the porous/disordered nature of the electrode and the presence of reversible solid-state Faradaic reactions resulting in pseudocapacitive behavior (e.g., pseudocapacitors, PCs). 

The use of electrochemical impedance spectroscopy (EIS) can be quite helpful for the study of complex electrode materials since some fundamental processes observed in the frequency domain cannot be accessed using the other electrochemical techniques [1]. Commonly, complex (e.g., porous/disordered) electrode materials exhibit distributed capacitance in the time and frequency domains due to hierarchically interconnected structural defects, which is usually modeled by an assembly of identical cylindrical pores (e.g., De Levie’s model). Alternatively, different electrode materials can be studied using a “macro homogeneous description of two closely mixed phases” to account for the “disordered behavior” foreseen from different nanostructured materials (e.g., the Paasch et al. and Bisquert et al. models—see further discussion). Also, some interesting insights were recently reported regarding the influence of mass transport and ohmic resistances on capacitive behavior [13].

Unfortunately, in different literature reports, the study of porous/disordered electrodes, as is the case of those used in supercapacitors, is accomplished using the EIS technique based on ad hoc equivalent circuit analogs (ECs) containing the diffusive Warburg element (*W*), which fits the experimental data well but fails to provide a physical correspondence with the fundamental processes occurring at the electrode/electrolyte interface, as well as in the liquid and solid phases (e.g., the presence of anomalous transport characteristics) [1]. Modified versions of the Randles–Ershler circuit (e.g., *R*_Ω_(*C*_edl_[*R*_ct_*W*])***C***, *R*_Ω_(*C*_edl_[*R*_ct_*W**C***]), etc.) are commonly used, with the incorporation of an additional capacitive element (in bold) to simulate the impedance response of a blocked electrode, as is the case with supercapacitors (SC). A typical misuse of irrational equivalent circuit analogs using the Warburg element in the absence of a charge transfer resistance was recently found in the literature [14] when a circuit containing only serial elements (e.g., *L*–*R*_o_–*Z*_W_) was used. This is nonsense since the Warburg impedance must always appear in series with the charge transfer resistance resulting in the Faradaic impedance (*Z*_F_), which appears in parallel to the electrical double-layer capacitance [15]. Unfortunately, several speculative analyses using the EIS can be found in the literature [16]. This critical question was recently discussed by some of the present authors [1]. 

The main source of confusion in the EIS reports involving the diffusive elements comes from the mathematical equivalence existing between the distinct theoretical models proposed for studying the porous electrode behavior (e.g., De Levie’s model for blocked interfaces where charge transfer is absent) and the diffusive mass transport coupled to Faradaic reactions (e.g., the Warburg and Kruger models for unblocked interfaces with a net charge transfer). These different models *predict the same impedance behavior* characterized by a phase angle of ≈ −45° since a mathematical equivalence arises from the general solution presented for second-order partial differential equations. 

Physical interpretations of complex-plane (Nyquist) plots for EDLCs were recently presented by Mei et al. [17] using simulations accomplished based on the modified Poisson–Nernst–Planck (MPNP) model for binary and symmetric electrolytes. Good qualitative findings were achieved by these authors without the use of conventional circuit models. However, the porous nature of real EDLCs was not considered in the theoretical model. Even so, the impedance profiles obtained by these authors are practically the same as those simulated using other robust impedance models [18,19,20,21]. Therefore, choosing the appropriate theoretical impedance model is a great challenge in EIS studies. 

Transmission lines (TLs) are special classes of equivalent circuits formally derived from fundamental laws according to particular transfer functions (TFs) representing the intrinsic properties of the studied system. Only after the pioneering TL model proposed by De Levie in 1963 [22] was a plausible physical meaning proposed for the dispersive capacitive effects verified at high frequencies for porous electrodes and characterized by a phase angle (*𝜑*) close to −45°. In this case, the accepted hypothesis is the dependency of the penetration depth of the sinusoidal wave inside the narrow pores with the applied frequency instead of a normal (Fickian) diffusive effect intrinsically coupled to a Faradaic reaction.

Outstanding contributions to the study of porous electrodes using the EIS were made in the last three decades by Lasia [18], Paasch et al. [19], Srinivasan and Weidner [20], Bisquert et al. [21], and other prominent authors [23,24,25]. A general literature survey of the impedance response of porous electrodes was recently reported by Huang et al. [26]. In particular, Bisquert et al. [21] proposed several impedance models to study different electrode materials considering the anomalous transport for the charge carriers in disordered media (e.g., semiconductors, some mixed oxides, and conducting polymers).

From the above considerations, we applied in this study a robust and generic impedance model for the study of complex electrochemical systems, which represents an effective macro homogeneous description of two closely mixed phases (Paasch–Micka–Gersdorf model) and provides a reliable description of the different physicochemical events occurring during the charge–discharge processes in porous electrodes used in SCs. The model contains two time constants (*τ*) representing the different events occurring in the liquid (Z_1_) and solid (Z_2_) phases, and a lumped impedance (*Z*_3_) describing the charge storage process at the solid/liquid interface. As a result, we can identify fundamentally different events occurring in supercapacitors. It is worth mentioning that the formalism introduced to the impedance model proposed by Bisquert et al. [21] and used in this work does not depend on the particular pore geometry, as is the case with the De Levie’s model [18], where evenly distributed identical cylinders compose the pores. It is worth mentioning that the influence of the physicochemical properties of porous electrode materials used in supercapacitors (e.g., intrinsic conductivities of the solid and liquid phases) was comprehensively incorporated by Srinivasan and Weidner in their well-known impedance model available in ZView^®^ software (Scribner Associates Inc., Southern Pines, USA), coded as DX-Type #8 [19]. However, the latter impedance model has a drawback since it demands previous knowledge of different properties obtained in independent (complementary) studies. In addition, various combinations of the intrinsic materials’ conductivity and capacitances can produce similar impedance spectra, leading to simulation findings that are difficult to interpret.

To the best of our knowledge, this is the first time that the Paasch–Micka–Gersdorf model has been used to study symmetric coin cells containing porous electrodes in the presence and absence of pronounced pseudocapacitance, considering the influence of the abnormal transport of the ionic and electronic charges. Using carbon nanofibers (CNF) and composite (CNF@NiO) electrodes housed in symmetric cells, we aim to instigate the charge transport in the electronic and ionic phases in intimate contact. At least in principle, the impedance model discussed in this work can be applied to different porous electrode materials used in SCs since the intrinsic chemical properties related to the electrode material are not incorporated in the present impedance model, i.e., only the structural material characteristics affecting the charge transport and frequency (capacitance) dispersion are relevant in the current context. Additional experiments were accomplished using cyclic voltammetry (CV) and galvanostatic charge–discharge (GCD) techniques to verify the internal consistency of the experimental findings.

## 2. Experimental Details

### 2.1. Preparation of the Composite Electrodes

The composite electrodes (model PC system) used in the symmetric coin cells consisted of carbon nanofibers grown by chemical vapor deposition on a porous carbon fabric substrate [27] and decorated with NiO particles using the incipient wetness impregnation method [8]. HexForce 3K carbon fabric fiber (Hexcel Co., Stamford, USA), composed of bundles containing carbon fibers with an average diameter of 7.5 µm, was applied as the current collector, which we used to grow carbon nanofibers. The pristine carbon fabric was cut into 5 cm × 3 cm pieces (e.g., ~1.0 g) and cleaned using a 63% (*w*/*w*) HNO_3_ solution for 1.0 h to remove the residues from the fabrication process. After drying in air at 110 °C for 12 h, the material was immersed in a 10 g dm^−3^ Ni(NO_3_)_2_⋅6H_2_O (Acros Co., New Jersey, USA), purity of 98%, alcoholic solution (e.g., 50% (*v*/*v*) ethanol-water) and subsequently dried for 12 h at 110 °C. Afterward, the fabric samples impregnated with nickel were calcinated in a pre-heated oven at 350 °C for 1.0 h using a quartz tube reactor where the samples were inserted and purged with argon gas at a volumetric flow rate of 100 cm^3^ min^−1^ for 10 min. In the sequence, under reductive atmospheric conditions to induce the formation of NiO nanoparticles, using a hydrogen flow rate of 200 cm^3^ min^−1^, the temperature inside the quartz tube was increased by 10 °C min^−1^ until it reached 400 °C. After 1.0 h at this temperature, to foster the correct conditions for the growth of the carbon nanostructures (nanofibers) using an ethane flow rate of 50 cm^3^ min^−1^ (e.g., the carbon source), the temperature inside the quartz tube was increased by 10 °C min^−1^ until it reached a final value of 700 °C, where it was held for 30 min. As a result, the nanofibers (e.g., CNFs) were grown on the surface of the carbon fabric substrate. After cooling the as-prepared material using an inert argon atmosphere, 14 mm diameter discs were cut from the modified carbon fabric and the samples were again impregnated with a 0.1 mol dm^−3^ Ni(NO_3_)_2_ alcoholic (e.g., 50% (*v*/*v*) ethanol–water) solution for 48 h. After immersion, samples were immersed in deionized water. After drying at 80 °C for 2 h, the discs were subjected to thermal treatment in a pre-heated oven at 350 °C for 2 h in an ambient atmosphere for the formation of NiO nanoparticles on the carbon nanofibers (e.g., NiO@CNFs). We will henceforward refer to porous carbon-based materials as CNFs and the composite electrode material as NiO@CNFs.

The overall masses of the CNF (model EDLC system) and NiO@CNF (model PC system) electrodes housed in the different symmetric coin cells were 12.3 and 12.6 mg, respectively. The specific capacitances reported in the electrochemical study were calculated using these masses.

### 2.2. Structural and Surface Morphology Characterization Studies

The surface morphology of the samples was examined using a FEI Inspect F-50 (Thermo Fisher Scientific, Hillsboro, USA) at 20 kV and with ETD detector. High-resolution images were obtained using a model 2100 MSC high-resolution transmission electron microscope (JEOL Inc., Peabody, USA). Samples were dispersed in isopropanol using an ultrasound bath and dropped on a TEM lacey carbon film fixed on the copper mesh. 

Raman spectra of the samples were performed on a Renishaw inVia spectrometer (Michigan, USA) using a 488 nm (Argon ion) laser, with an integration time of 60 s and a 50× LWD objective lens in the range of 100 to 2500 cm^−1^. The spectra analyses were accomplished by proper subtraction of the baseline signal. At the same time, the curve fitting procedure was performed in the region from 300 to 1750 cm^−1^ using the Lorentzian and Gaussians functions available in the software Fityk (Open Free).

The surface chemistry of the composite samples was analyzed using the K-alpha radiation with the aid of a Thermo Scientific (Massachusetts, USA) X-ray photoelectron spectrometer.

The structure of the composite material was characterized by X-ray diffraction analysis, performed with a model PAN analytical X’Pert PRO X-ray diffractometer (Malvern, UK) using Co-K*α* radiation (*λ* = 0.178901 nm) in a Bragg–Brentano *θ*/2*θ* configuration (Goniometer PW3050/65). The diffraction patterns were collected at steps of 0.04° and the accumulation time of 5 s per step within the 2*θ*-scale range from 20° to 80°.

The specific surface area (SSA/m^2^ g^−1^) obtained according to the BET method was measured using nitrogen at 77 K with a Micromeritics ASAP 2010 instrument (Norcross, USA). Before measuring, 100 mg of the active powder carbon fiber material was degassed at 100 °C for 12 h. Finally, the powder sample was conditioned at 200 °C to obtain a constant pressure of 0.02 μm Hg.

### 2.3. Electrochemical Characterization Studies

All electrochemical experiments were performed using a model CR2032 coin cell in the symmetric configuration containing a cellulosic filter paper soaked with a 1.0 mol dm^−3^ Li_2_SO_4_ aqueous solution to avoid short circuiting and to provide ionic conduction. A 302N potentiostat–galvanostat with an FRA module from AUTOLAB^®^ (Utrecht, The Netherlands) was used throughout.

The study in the frequency domain was accomplished using the EIS technique, whereby a fixed cell potential corresponding to the open circuit cell potential (OCCP) was applied while the frequency of the superimposed alternated cell potential was swept from 100 kHz to 10 mHz using a low sinusoidal signal of 10 mV (peak-to-peak). The quantitative analysis of the impedance data was carried out based on the double-channel transmission line model denoted as Bisquert #2 using NOVA^®^ software from AUTOLAB^®^ (Utrecht, The Netherlands), where the fitting/simulation procedure was conducted using the complex nonlinear least squares (CNLS) method. Very good simulations were obtained in all cases (*r*^2^ ≥ 0.998), with a relative error for each model’s parameter of less than 2%. Please see further details in the discussion section.

Cyclic voltammetry (CV) and galvanostatic charge–discharge (GCD) experiments were also performed using the symmetric coin cells. Voltammetric curves (VCs) were obtained at different scan rates (e.g., 5, 25, and 50 mV s^−1^) for the pseudocapacitive voltage range of 1.0 V, while the GCD curves were measured at 4 A g^−1^. The specific capacitance (*C*/F g^−1^) was also determined from the galvanostatic charge–discharge using the equation *C* = *I*/*m*(Δ*V*/Δ*t*), where *I* is the negative (cathodic) current, *m* is the overall electrode mass (cathode and anode), and Δ*V*/Δ*t* is the negative slope of the discharge curve.

## 3. Results and Discussion

As already emphasized, our intention with this work is to obtain electrochemical information concerning the fundamental events during the charge storage process, i.e., charge transport anomalies involving the ionic and/or electronic charge carriers, which are affected by the disordered electrode materials used in EDLC and PC symmetric devices. However, to provide further information about the synthesized model materials, we present in this section an ex situ characterization. 

### 3.1. Ex Situ Characterization Studies

#### 3.1.1. Surface Morphology Analysis: SEM and TEM Studies

Figure 1 shows SEM and TEM micrographs of the CNF (Figure 1a–c) and NiO@CNF (Figure 1d–f) materials. Figure 1a,d presents SEM images of the carbon fabric macrostructure before (a) and after (d) the presence of the as-grown CNFs. Obviously, at this magnification, no significant changes in the surface morphology can be detected due to the presence of CNFs. Figure 1b shows an image with higher magnification, where we can verify the formation of CNFs with a spaghetti-like structure, i.e., CNFs were grown on fabric fibers in an entangled way. After immersion of the CNFs into the nickel nitrate aqueous solution, the fibers of the CNF tend to join together, forming a sponge-like electrode surface (see Figure 1e). Figure 1d–f shows SEM and TEM images of the CNFs’ structures after decoration with NiO nanoparticles to obtain the composite electrode material (e.g., NiO@CNFs). As seen, TEM micrographs (Figure 1c,f) evidenced the presence of NiO on the CNF surface.

As verified from the TEM analysis, the modified CNFs are highly defective, with a nonlinear structure. The CNFs diameter range from 10 to 90 nm. At the same time, NiO exhibited quasispherical nanoparticles with diameters ranging from 1 to 5 nm.

#### 3.1.2. Raman and XPS Studies

Raman spectra of CNFs and NiO@CNFs are presented in Figure 2. Raman spectra of CNFs (Figure 2a) were deconvoluted into four peaks referring to the G and D bands: G (~1600 cm^−1^), D (~1360 cm^−1^), D1 (~1270 cm^−1^), and D2 (~1530 cm^−1^). G and D bands were assigned to the *sp*^2^ in-plane carbon vibration and out-of-plane vibrational modes, respectively [28]. D1 and D2 bands were ascribed to the C-C and C=C stretching modes (e.g., *sp*^3^ and *sp*^2^ bonds) and to the amorphous carbons containing different organic species, respectively [29]. Figure 2b shows the Raman spectrum of the NiO@CNFs composite, which was deconvoluted into seven peaks. The spectrum of NiO@CNFs also contains three additional characteristic bands due to the presence of NiO. The bands were located at ~370 cm^−1^, ~498 cm^−1^, and ~740 cm^−1^. The 498 cm^−1^ peak is assigned to a lack of symmetry, i.e., defects due to high nickel vacancy concentration, around the atoms normally participating in the formation of phonons in a perfect crystal [30,31]. The 370 cm^−1^ peak is attributed to TO, and 740 cm^−1^ is harmonic 2TO modes [30,31,32,33]. From this analysis, we concluded that nickel oxide nanoparticles are highly defective due to the intensity of the 498 cm^−1^ peak. 

A comparison of the XPS findings obtained for the two different samples was performed. In this sense, Figure 3 shows the long-range XPS spectra (Figure 3a,d), as well as the short-range spectra for the different chemical species: C1s (Figure 3b,e), O1s (Figure 3c,f), and Ni2p (Figure 3g). 

Different deconvolutions were accomplished using Gaussian and Lorentzian functions with the Shirley baseline correction. Figure 3a,d shows the individual long-range region spectrum for the different materials. It is possible to verify the presence of the Ni2p peak for the NiO@CNFs, while it is absent for the CNFs, as expected. Although the CVD synthesis of CNFs uses nickel as the catalyst, its concentration in the sample is too low to be detected. This analysis is consistent with the Raman findings (see Figure 2). Figure 3b,c shows the main peak at ~284.8 eV, attributed to the C=C bond, which confirmed the presence of sigma and *π* bonds characteristic of the graphene structures present in the CNFs [34]. The C=O and C−C=O bonds, and *π−π** transitions, were verified for the two distinct samples [35]. However, the C-OH bond was only verified for the CNFs and C-O-C only for NiO@CNFs. 

The C1s region of the spectrum showed minor changes in the carbon structure due to the presence of nickel oxide nanoparticles, as previously observed from the TEM analysis (see Figure 1). On the contrary, the O1s region of the spectrum revealed a considerable increase in the XPS signal from 2 to 8 at. % after the decoration of CNFs with NiO. To identify the presence of Ni–O, the XPS spectra of the O1s and Ni2p regions were used. In the O1s region, the spectra were deconvoluted into two bands for the CNFs sample. The presence of C=O, C-O, and C-OH bonds was confirmed for the C1s region [36]. 

The formation of NiO nanoparticles occurred after the annealing process of CNFs, carried out at 350 °C in the presence of oxygen and Ni^2+^ species, as confirmed by the C−O and Ni−O bonds found in the XPS spectra. It is worth mentioning that the presence of the H−O−H bond in the XPS spectrum comes from air humidity. At the same time, the presence of the C=O bond was assigned to the carbonyl bonds, while the occurrence of the CO_3_^2^^−^ was ascribed to the carboxylic groups, and Ni–O is due to the oxide formation.

Figure 3g shows the Ni2p region of the XPS spectrum obtained for the NiO@CNFs composite. Even considering that the amount of nickel oxide nanoparticles present in the carbon structure is small, it was possible to obtain a high-resolution Ni2p spectrum to observe the Ni2p_(3/2)_ and Ni2p_(1/2)_ regions. The Ni2p_(3/2)_ region was composed of two peaks and it was possible to observe two oxidation states for nickel (e.g., Ni^2+^ and Ni^3+^), i.e., we observed the formation of NiO and Ni(OH)_2_ species (e.g., Ni^2+^ comes from the multiplet splitting referring to NiO) [37]. The Ni2p_(1/2)_ region is a doublet of the Ni2p_(3/2)_ region, so we have the same peaks. These data are in agreement with the Raman findings already discussed in this work (see Figure 2b). All energies (B.E.) referring to different chemical bonds are shown in Table 1.

#### 3.1.3. XRD and BET Studies

We performed X-ray diffraction (XRD) on CNFs and NiO@CNFs to characterize the crystalline structure. Figure 4 shows the XRD patterns obtained.

The analysis of Figure 4a confirms the existence of carbon (*sp*^2^) structures (PDF# 01-089-8487), with the peaks corresponding to the (0 0 2) and (1 0 1) *hkl*-planes. In Figure 4b the peaks correspond to two different crystalline phases, i.e., the *sp*^2^ carbon structure and NiO (cubic crystal system and the *Fm-3m* spatial group), in agreement with the (1 1 1) and (2 0 0) *hkl*-planes (PDF# 00-001-1239). These findings agree with the XPS, and Raman analyzes and confirm the presence of carbon and NiO phases. The presence of the (2 0 0) *hkl*-plane corroborates the interplanar spacing of 0.208 nm in the TEM study. 

Figure 5a,c shows the volume of N_2_ corresponding to the adsorption/desorption processes as a function of the relative pressure used to obtain the isotherms according to the BET analysis, while Figure 5b,d shows the incremental pore size distribution calculated using the BJH model.

The N_2_ adsorption/desorption isotherms are shown in Figure 5a. The isotherm obtained for the CNFs exhibited an IV type according to the IUPAC classification; there is a region (e.g., ~0.6 to 1.0 *P*/*P*_0_) where the relative pressure slightly varied when the adsorbed volume increased. These isotherms indicated the presence of slit-type pores. Due to the capillary condensation process, there was a hysteresis loop in the range of ~0.6 to 1.0 *P*/*P*_0_ that can be classified as H3 type, i.e., in this case, we have complete filling of the mesopores at relative pressures lower than 1.0 *P*/*P*_0_. The profile showed in Figure 5a was mainly characterized by two relatively vertical asymptotic branches at *P*/*P*_0_ = 1, which are associated with a nonrigid aggregation of the plate-shaped particles, giving rise to slit pores, thus showing a higher specific surface area for CNFs of ~53 m^2^ g^−1^ (see Figure 5a) [38]. The pore size distribution (see Figure 5b inset) revealed the presence of micropores (*d* < 2.0 nm), mesopores (2 nm < *d* < 50 nm), and macropores (*d* > 50 nm) [39]. Therefore, it can be concluded that the largest contribution to the total specific surface area is due to mesopores. 

Accordingly, the corresponding value obtained from the NiO@CNF sample was ~41 m^2^ g^−1^ (see Figure 5c). From these findings, we can affirm that some pores of the carbon structure were clogged during the oxide formation, thus reducing the specific surface area compared to the CNFs sample. As seen in the case of the NiO@CNF sample, the major contribution to the specific surface area is due to mesopores (Figure 5c). The isotherm obtained for the composite material exhibited an IV type with a characteristic hysteresis loop in the range of ~0.8 to 1.0 *P*/*P*_0_ of the H3 type.

### 3.2. Frequency Domain Analysis Using the EIS Technique: Identification of Charge Transport Anomalies during the Charge Storage Process in EDLC and PC Devices

Figure 6 shows the complex plane (Nyquist) plots obtained for different symmetric coin cells containing the (a) CNF or (b) NiO@CNF electrodes, and the (c) generic double-channel transmission line model, where each impedance *Z* is composed of a circuit containing an ohmic resistor (*R*) in parallel to a constant phase element (CPE) representing the different dispersive effects. As can be seen, the impedance response was characterized by the following characteristics: (i) the presence of a resistive/capacitive arc in the high-frequency range; (ii) the occurrence of an inclined line with a phase angle close to −45° at medium frequencies, and (iii) the presence of an almost vertical straight line in the low-frequency range with a phase angle close to −90°. According to [19], this type of impedance behavior, in the absence of irreversible Faradaic reactions (e.g., water splitting), is in agreement with the theoretical predictions of the porous electrode model, with the additional inclusion of the anomalous transport phenomenon for the ionic and/or electronic charge carriers. The anomalous transport effects occur from the medium- to the high-frequency interval of the impedance spectrum [21].

More specifically, when the intermixed solid (*Z*_1_) and liquid (*Z*_2_) phases have a similar *dc* resistance, the influence of the disordered solid phase on the electronic transport commonly appears as an arc in the high-frequency region of the impedance spectrum (see Figure 6a,b). However, it is worth mentioning that the *i*nterfacial (parallel) impedance (*Z*_3_), including the electrical double-layer capacitance and pseudocapacitive effects, can also affect this region of the spectrum.

Inspired by the classical solid-state physics studies regarding the anomalous charge transport in disordered solids, Bisquert et al. [21] proposed a robust generic impedance model for the study of different types of electrochemical systems with disorder/dispersive characteristics (e.g., a power law behavior or dispersive effects). In addition, Bisquert [40] also discussed some particular cases of great interest for studying the different electrochemical systems used in technological applications. The total impedance (*Z*_total_) for the porous or mixed-phase electrodes, considering the presence of anomalous transport and the additional influence of the uncompensated ohmic resistance (*R*_ESR_) intrinsic to the supercapacitors, is given by the following transfer function [21]: (1)$$Z_{\mathrm{total}}=R_{\mathrm{ESR}}+\frac{Z_{1}Z_{2}}{Z_{1}+Z_{2}}\left[L+\frac{2\lambda }{\sin h\left(\frac{L}{\lambda }\right)}\right]+\lambda \left(\frac{Z_{1}{}^{2}+Z_{2}{}^{2}}{Z_{1}+Z_{2}}\right)\coth\left(\frac{L}{\lambda }\right),$$
where: (2)$$\lambda ={\left(\frac{Z_{3}}{Z_{1}+Z_{2}}\right)}^{1/2}$$
(3)$$Z_{1}=\frac{r_{1}}{1+r_{1}q_{1}{\left(j\omega \right)}^{n}}$$
(4)$$Z_{2}=\frac{r_{2}}{1+r_{2}q_{2}{\left(j\omega \right)}^{s}}$$
(5)$$Z_{3}=\frac{1}{q_{3}{\left(j\omega \right)}^{\beta }}.$$

The parameter *L* is the length of the intermixed (porous) phase, and is commonly unknown. From the point of view of the macro aspects of the impedance model (see further discussion in this section), the particular *L*-value is irrelevant for most applications. However, to obtain internal consistency between the unities of the transverse and parallel impedances, the apparent length of the porous electrode layer is given in the current work by the equation *L* = *m*/*ρA*, where *m* is the mass of the active layer, *ρ* is the apparent density of the electrode layer, and *A* is the geometric surface area of the porous electrode.

The individual (macroscopic) parameters measured are *R*_1_ = *Lr*_1_ (e.g., the total *dc* electrolyte resistance inside the irregular ionic channels), *Q*_1_ = *q*_1_/*L* (e.g., the information about the anomalous ionic transport mechanism or the transversal electrolyte capacitive-like effects), *R*_2_ = *Lr*_2_ (e.g., the total *dc* electrode resistance), *Q*_2_ = *q*_2_/*L* (e.g., the anomalous electronic transport in the electrode material or the transverse electrode capacitance), and *Q*_3_ = *q*_3_*L* (e.g., the constant phase element coefficient (*Q*_edl_*), which is representative of the nonideal overall electrical double layer, including pseudocapacitance). As emphasized by Bisquert et al. [21], impedances *Z*_1_ and *Z*_2_ represent “single transport mechanisms” rather than a conventional association of resistive and capacitive elements used in purely electrostatic processes. 

In light of the theoretical analysis proposed by Paasch et al. [19], the impedance model represented by Equation (1) agrees with the effective macro homogeneous description of two closely mixed phases. The latter is more appropriate for practical applications than the traditional view, where the porous electrode model is derived based on the presence of a perfect cylinder geometry whose pore length is longer than its diameter. As it is a common practice in EIS studies to facilitate the numerical analysis of the experimental findings, the transfer function described by Equation (1) can be represented by the equivalent double-channel transmission line shown in Figure 6c. In this model, *Z*_1_ and *Z*_2_ are impedances per unit length (Ω m^−1^) transverse to the macroscopic outer surfaces, while *Z*_3_ is the impedance length (Ω m) parallel to the macroscopic surfaces. The impedance *Z*_3_ represents the capacitance or pseudocapacitance of the electrode/solution interface, i.e., *Z*_3_ = 1/*Q*_3_(*j**ω*)*^β^*. For convenience, the transverse and parallel impedances are represented using gravimetric quantities (e.g., *Z*_1_/Ω g^−1^, *Z*_2_/Ω g^−1^, and *Z*_3_/Ω g), since, as discussed above, *L* ∝ *m*.

The anomalous charge transport across narrow pores filled with the electrolyte can be understood considering the impedances *Z*_1_ and/or *Z*_2_ must be frequency-dependent, exhibiting a power law behavior (e.g., *n* < 1 and/or *s* < 1; see Equations (3) and (4)). These particular impedances must be lower at high frequencies than at lower frequencies due to the effect of narrow ‘throats’ and bottleneck structures that restrict the long-range (e.g., low-frequency) ionic motion, as well as the presence of disordered structures in the solid phase, which affects the electronic transport [21]. In general, the dispersive effects commonly found for semiconductors and conducting polymers and represented by the power law expressions (see Equations (3) and (4)) due to the anomalous charge transport in disordered phases being associated with discrete transitions between localized states that originated from the different types of structural disorder/defects present in porous electrode materials, which can be characterized by a characteristic time constant or relaxation time (*τ*_c_) [41].

To overcome the difficulties involving the specification of a particular conduction mechanism, Bisquert et al. [21] considered the problem of *anomalous charge transport* using the concept of a model-independent macroscopic phenomenological theory based on a generalized constitutive equation and the general Einstein relationship for diffusional processes. As a result, different types of electrode materials that exhibit dispersive effects for charge transport can be studied using the proposed model. In this sense, the porous electrodes operate by simultaneous transport of electronic and ionic species occurring in the solid and liquid phases, respectively, to attain the principle of local electroneutrality. In this scenario, the solid phase provides a continuous path for the transport of electrons, but the dimensions of the structural elements in the disordered solid are quite small, especially in the presence of nano-sized domains. Accordingly, the electrolyte penetrates the accessible void regions present in the solid phase, resulting in very narrow liquid channels that exhibit a high degree of tortuosity. As a result, the electrode system is characterized by two closely mixed phases with a possible degree of disorder for the electronic and ionic charge carriers [21].

The results of the CNLS simulation are summarized in Table 2. As can be seen, the decoration of the CNFs with NiO caused strong changes in the electrochemical characteristics of the composite electrode. The major impedance parameters accounting for the performance of supercapacitors are *R*_ESR_ and *Q*_edl_. As seen, the *R*_ESR_ value decreased after the decoration process, which is very good for the overall performance of the coin cell.

In principle, these findings indicate that the spontaneous hydration of the nickel oxide (e.g., NiO + H_2_O → Ni(OH)_2_) promoted an increase of the wettability of the inner surface regions of the porous electrode, thus increasing the overall cell conductivity. As seen, the apparent (fractal) electrical double-layer capacitance verified for the carbon-based scaffold or the apparent (fractal) pseudocapacitance observed for the decorated electrode was characterized by a dispersive exponent very close to 1 (e.g., *β* ≥ 0.95). This implies with great accuracy that *Q* ≈ *C*. It was verified that the decoration process using NiO strongly increased the coin cell charge storage characteristics from 2.62 to ≈ 536 F g^−1^ (since *β* ≈ 1), promoting an enormous improvement in the device’s performance. These findings explicitly reveal the paramount importance of the reversible solid-state redox reactions (RSR) to the overall charge storage process.

Considering the lower specific surface area exhibited by the composite material compared to the carbon-based scaffold, one can propose that the main contribution to the overall specific capacitance is a bulk-like or near-surface pseudocapacitance instead of the purely surface electrostatic process. We can argue that the strong electrochemical activity exhibited by Ni(OH)_2_ for the charge storage process is due to the presence of RSR involving the Ni^2+^/Ni^3+^ redox couple [42]. According to the literature, the redox reaction involving Ni(OH)_2_ is a solid-to-solid transformation [43]. We present the well-known Bode’s reaction scheme to represent the electrochemical behavior of Ni(OH)_2_ during the charging–discharging process [44]:*α*-Ni(OH)_2_ → *γ*-NiOOH + H^+^ + e^−^(6)
*β*-Ni(OH)_2_ → *β*-NiOOH + H^+^ + e^−^
*α*-Ni(OH)_2_ → *β*-Ni(OH)_2_
*γ*-NiOOH → *β*-NiOOH.

In short, the *α* ↔ *γ* and *β* ↔ *β* pathways, involving the solid-state redox transitions of the type Ni^2+^/Ni^3+^, are of interest to the reversible charging–discharging process in supercapacitors. The *α* → *β* phase change only occurs under potentially extended cycling, while the *γ* → *β* transformation occurs at the expense of excessive charge insertion.

For convenience, the gradual changes in the Ni(OH)_2_ material occurring during proton intercalation/deintercalation can be simplified as follows [45]:Ni(OH)_2_ ↔ NiO*_x_*(OH)_(2 − *x*)_ + *x*H^+^ + *x*e^−^,(7)
where it is commonly assumed that the *β* ↔ *β* pathway of the Bode’s scheme is dominant. It is worth mentioning that, due to the nature of amphoteric oxide, the issue of whether the species that diffuses through the Ni(OH)_2_ structure is H^+^ and/or OH^−^ remains open to discussion [46]. 

Another possible origin for the pseudocapacitive behavior involving Ni(OH)_2_ might be the intercalation/deintercalation of Li^+^ ions into the hydrated oxide structure. According to [45], the intercalation/deintercalation of Li^+^ in Ni(OH)_2_ is more probable in a concentrated LiOH aqueous solution or in rigorously anhydrous electrolytes (e.g., LiClO_4_ + PC (propylene carbonate)). According to Faria et al. [47], who proposed the occurrence of an exchange reaction, when different cations are present in the solution the proton intercalation can govern the overall charging–discharging processes comprising a mixed intercalation process, involving to a minor extent the intercalation/deintercalation of species like Li^+^. Thus, it is plausible to consider for the hydrated oxide the following parallel process involving a substitutive solid-state redox reaction:Ni(OH)O*_x_*H*_y_* + *z*Li^+^ ↔ Ni(OH)O*_x_*Li_(*y*_
_− *g*)_ + *g*H^+^ + e^−^.(8)

Therefore, it is prudent to consider for aqueous solutions that the overall pseudocapacitive process is a combined process given by the above solid-state reactions (Equations (7) and (8)) [46].

The other set of impedance parameters presented in Table 2 involves the ionic and electronic transport characteristics in the disordered phases. The ionic resistance inside the irregular pores decreased about 5-fold in the presence of Ni(OH)_2_, while the exponent (*n*) representing the anomaly degree increased from 0.42 to 0.92, i.e., the anomalous ionic transport practically disappeared for the composite electrode. In addition, we found for the decorated electrode that *R*_ionic(CFs)_ > *R*_ionic(NiO@CNFs)_. In principle, these findings suggest the occurrence of a Grotthuss-like mechanism for protons and/or Li^+^ species inside the hydrated (gel layer) structure that formed during hydration of the nickel oxide. As a result, there is the formation of more regular paths for the ionic transport during the charge–discharge events and, therefore, the overall ionic transport in both liquid and solid (gel layer) phases behaves similarly to a normal (Fickian) process. On the contrary, for the carbon-based electrode, the transport of protons and/or Li^+^ species is abruptly interrupted at the blocked electrode/solution interface, resulting in a severe anomaly during the fast charge–discharge process. Also, the more disordered structure of the carbon material creates zig-zag paths that decrease the mean free paths in narrow channels, thus reducing the overall ionic conductivity.

According to [48], the diffusion coefficients for protons (*D*_H_^+^) in the bulk liquid phase and the hydrated nickel oxide (*D*_H_^+^_(NiO)_) are 9.32 × 10^−5^ cm^2^ s^−1^ and ≈3 × 10^−10^ cm^2^ s^−1^, respectively. Therefore, at least in principle, the Faradaic intercalation reaction during the charge storage process can be controlled by the slower ionic transport on the solid phase, which is sometimes indirectly verified in several *battery-like* systems by the irreversible peaked shape voltammograms obtained at low scan rates (e.g., *ν* ≤ 10 mV s^−1^) and/or the pronounced voltage plateau verified in the GCD curves [49]. It was verified in the current work that *n*_(NiO)_ = 0.92 for ionic transport in the hydrated composite electrode and, therefore, we can consider that the overall proton diffusion in the two closely mixed phases is practically a nondispersive (Fickian) event. Thus, in the presence of a concentrated supporting electrolyte to sustain the migration current, there is the general phenomenological relationship *J* = *D*(Δ*C*_H_^+^/Δ*x*) for Nernst’s layer, where Δ*C*_H_^+^/Δ*x* is the concentration gradient driving the flow (*J*), and *D*/Δ*x* = *k*_mt_ is the diffusion mass transport coefficient. In addition, from the continuity condition, we know that the three different coupled flows involved in the Faradaic (pseudocapacitive) process must be equal (e.g., *J*_(e_^−^_)_ = *J*_(H_^+^_)_ = *J*_H_^+^_(NiO)_). Therefore, considering that *D*_H_^+^ >> *D*_H_^+^_(NiO)_, we propose that *k*_(mt)-H_^+^_(NiO)_ is not so different from *k*_(mt)-(H_^+^_)_, i.e., as already mentioned, the composite electrode material can provide many alternative routes for the ionic and electronic charges, thus resulting in short-range paths (Δ*x*) for proton transport. As a result, the flow of ionic charge considerably increases in the hydrated oxide structure. The electrochemical system behaves like a real capacitive system instead of a battery-like one.

Sharma et al. [50] recently reported on tuning the nanoparticle interfacial properties and stability of the core shell structure in Zn-doped NiMoO_4_@AWO_4_ electrodes. They considered Zn-doped nickel molybdate (NiMoO_4_) (ZNM) as a core crystal structure and AWO_4_ (A = Co or Mg) as a shell surface. They verified the ability to tune the core shell nanocomposites with surface reconstruction as a source for surface energy (de)stabilization. It was verified that the performance of the core shell is significantly affected by the chosen intrinsic properties of metal oxides with high performance compared to a single-component system in supercapacitors. The constructed asymmetric device (e.g., Zn-doped NiMoO_4_@CoWO_4_ (ZNM@CW)||activated carbon) exhibited superior pseudocapacitance, delivering a high areal capacitance of 892 mF cm^−2^ at 2 mA cm^−2^ and excellent cycling stability (i.e., 96% capacitance retention after 1000 charge–discharge cycles). Sharma et al. [50] presented theoretical and experimental insights into the extent of the surface reconstruction to explain the storage properties in SCs.

The diffuse or drift effects in a disordered solid phase containing many traps are correlated with the number of electrons effectively contributing to the charge flux, which in turn depends on the frequency. Thus, the presence of dispersive (anomalous) effects can be identified through the dispersive exponent (*s*) in Equation (4) [51]. Using the Nernst–Einstein relationship, we can verify that, for most crystalline electronic conductors, the diffusion coefficient for the electrons (*D*_e_) is in the range of 7 to 241 cm^2^ s^−1^ [52,53]. However, in some cases involving disordered electronic materials where *s* << 1, the *D*_e_ values can be drastically reduced by up to two orders of magnitude due to the pronounced multiple trapping events caused by the structural inhomogeneities present in the solid phase. On the contrary, we observed a nearly 2-fold improvement in the electronic transport process after the electrode decoration with NiO. Since *s* ≥ 0.98, we verified that the anomalous transport in the solid electrode structure is practically absent for the different materials. These findings indicate that, in the nano-sized domains of the electrode material, the electronic transport behaves like a normal one. A comparison of the experimental findings revealed that *R*_ionic_/*R*_electronic_ = 16. This small discrepancy in the resistance values is responsible for the presence of a well-defined straight line in the complex plane plot (see Figure 6b) in the medium-frequency range with a phase angle close to −45°, as is theoretically predicted when *Z*_1_ ≈ *Z*_2_ [21]. By contrast, when this resistance ratio is very high (e.g., *R*_ionic_/*R*_electronic_ > 100), the transmission line behavior is dictated only by the high impedance channel, i.e., the double-channel transmission line is converted to a single-channel one, and the complex plane plot exhibits a different profile to that shown in Figure 6b.

The overall DC resistance (e.g., the purely resistive impedance) incurred by the normal and/or anomalous transports of the electronic and ionic charge carriers in the two channels of the transmission line representation can be determined from the low-frequency limit (*ω* → 0) of Equation (1) (the physical model) [54]:(9)$$Z_{d.c.}(\omega \to 0)=\frac{\chi_{1}\chi_{2}}{\chi_{1}+\chi_{2}}L=\frac{R_{\mathrm{ionic}}R_{\mathrm{electr}.}}{R_{\mathrm{ionic}}+R_{\mathrm{electr}.}}.$$

Thus, the impedance of the transmission line is characterized by two ohmic resistances in parallel. It is predicted that the almost vertical capacitive straight line verified at very low frequencies for blocked electrodes, as is the case with supercapacitors, can be displaced more or less to the right-hand side of the complex plane plot, depending on the magnitude of the ohmic resistances imposed by the liquid and solid phases in intimate contact. Obviously, if *R*_1_ >> *R*_2_, *Z*_d.c._ ≅ *R*_2_. On the other hand, *Z*_d.c._ ≅ *R*_1_. These conditions imply that a single ohmic resistor describes the overall DC resistance. However, this is not the case in the present study since we verified that *R*_1_ ≈ *R*_2_. We found that the *Z*_d.c._ values for the CNF and NiO@CNF electrodes were 203 Ω g^−1^ and 88 Ω g^−1^, respectively. These findings reveal that the dissipative effects caused by ohmic losses are lower for the decorated electrode.

The crossover frequencies (*ω*_c_) between DC and AC regimes for the transverse (e.g., *Z*_1_ and *Z*_2_) impedances of the two closely mixed phases are as follows [53]:(10)$$\omega_{c}(Z_{1})=\frac{1}{{\left(r_{1}q_{1}\right)}^{1/n}}=\frac{1}{{\left(R_{\mathrm{ionic}}Q_{\mathrm{ionic}}\right)}^{1/n}}$$
(11)$$\omega_{c}(Z_{2})=\frac{1}{{\left(r_{2}q_{2}\right)}^{1/s}}=\frac{1}{{\left(R_{\mathrm{electr}.}Q_{\mathrm{electr}.}\right)}^{1/s}}.$$

Thus, considering the relationship between the time constant (e.g., *τ*_c(1)_ = (*RQ*)^1/*n*^ and *τ*_c(2)_ = (*RQ*)^1/*s*^), and the characteristic frequency (*f*_c_) given by *f*_c_ = 1/*τ*_c_, one can evaluate these quantities for the ionic and electronic transport phenomena using Equations (10) and (11), respectively, as shown in Table 3.

The analysis of the data in Table 3 revealed two distinct scenarios for the different electrode materials used in the symmetric coin cells. In the case of the CNF electrodes, the time constants (*τ*_c_) were not so different for the distinct charge transport in the ionic (*τ*_c(1)_ = 1.25 s) and electronic (*τ*_c(2)_ = 0.67 s) phases. The very close characteristic frequencies of 0.81 s^−1^ and 1.50 s^−1^ revealed that the transition between DC and AC regimes for the ionic and electronic phases is coupled and occurs in the low to medium frequency range of the impedance spectrum. Therefore, the formation of a well-defined semicircle at the high frequencies characterized by a given time constant was not observed (see Figure 6a inset). For comparison, using a multiscale impedance model to study different supercapacitors using carbon-based porous electrodes, Huang et al. [14] recently reported *τ*_c(1)_ values in the range of 0.16 s to 2.35 s for their different electrodes. However, due to the limitations of the model, they did not evaluate the *τ*_c(2)_ values and the eventual presence of anomalous transport. 

The analysis of the other case comprising the NiO@CNF composite electrodes revealed very different time constants of *τ*_c(1)_ = 2.70 × 10^−4^ s and *τ*_c(2)_ = 2.06 s for the ionic and electronic charge transports, respectively. Thus, we obtained different characteristic (crossover) frequencies of 3.71 × 10^3^ s^−1^ and 0.48 s^−1^ representing the transition regions in the complex plane plot for the ionic and electronic transports, respectively. As a result, three different regions can be clearly identified in the impedance spectrum (see Figure 6b): (i) the well-defined semicircle at high frequencies; (ii) the false Warburg-like straight line localized in the medium–low-frequency region, and (iii) the capacitive/pseudocapacitive straight line at very low frequencies. It is worth noting that the physicochemical origin of the high-frequency semicircle in the case of porous blocked electrodes is not due to a Faradaic (charge transfer) reaction. On the contrary, this behavior, which is predicted by the theory represented by Equation (1), is due to a combination of two factors: (i) the penetration depth dependency of the sinusoidal perturbation on applied frequency, and (ii) the presence of an “abnormal charge transport” phenomenon involving the electronic and/or ionic charge carriers in disordered media, i.e., the different phases being in intimate contact along the porous electrode surface. Unfortunately, this vital aspect, which has been known about for a long time, is ignored in most literature reports dealing with SCs. Commonly, authors not acquainted with the theoretical aspects of EIS misuse equivalent circuit analogs without a physicochemical meaning. 

In fact, with rare exceptions, most of the circuit analogs used in EIS studies are merely statistical devices used as part of a trial and error method to obtain a good simulation of the impedance data using the different analog circuits present in commercial software packages. The impedance data are often simulated with good statistics. However, as expected, this nonphysical approach fails to provide a physicochemical meaning to the used circuit’s parameters. For more details, please see the excellent books by Barsoukov and Macdonald [54], Orazem and Tribollet [55], and Lasia [18], which constitute a trustworthy source of knowledge about the impedance method of analysis. Unfortunately, as can be seen in several papers, the current literature using the EIS technique for dealing with energy storage devices is full of fundamental errors that are propagated in several important articles published by different research groups.

A possible explanation for the large difference in the time constants verified for the electron transport in the different electrode materials (e.g., *τ*_c(carbon material)_ = 1.25 s and *τ*_c(composite)_ = 2.70 × 10^−4^ s) is the pronounced occurrence of the RSR involving the Ni^2+^/Ni^3+^ redox couple, i.e., due to the Faradaic nature of the charge storage process, a great fraction of the transported electrons taking part in the RSR process is subjected to an activation barrier, involving the presence of entropic (dissipative) effects, as predicted by the transition state theory (TST) [56]. As a result, the overall flux of electrons measured in the impedance experiments is partially suppressed by an energetic barrier in the case of the composite electrode, thus decreasing the time constant.

In general, the above discussion involving the EIS technique conveys the importance of using relevant physicochemical models represented by a transfer function (or the equivalent transmission line representation) for the detailed analysis of the significant events occurring during the charge storage process in SCs. Therefore, it becomes evident that the use of equivalent circuit analogs, commonly derived from modified versions of the well-known Randles–Ershler circuit, which was further improved by Grahame and Sluyters with the concept of Faradaic impedance for the study of different electrochemical reactions on the mercury electrode, must be used with great caution to avoid speculative discussions involving solid electrodes used in different energy storage devices that exhibit different dispersive effects.

### 3.3. CV and GCD Analyses

Obviously, there are several important parameters determined using EIS that are not available from the CV and GCD experiments. However, to verify the internal consistency of the specific capacitance experimental findings obtained using different electrochemical techniques, we also conducted studies using the CV and the GCD techniques. As will be seen, good agreement was found since the CV and GCD profiles are representative of true supercapacitors instead of undesirable battery-like systems. Also, the specific capacitance values determined from the EIS and GCD findings are similar. 

Figure 7 contrasts the CV and GCD findings obtained for CNFs and NiO@CNFs in two distinct symmetric coin cells filled with a 1.0 M Li_2_SO_4_ aqueous solution. As seen in Figure 7a, we obtained a capacitive/pseudocapacitive voltage window of 1.0 V for both cells. Two important features ensured that our coin cells behaved as true supercapacitors: (i) the voltammetric curves were almost rectangular, with a “mirror-like” shape even at 50 mV s^−1^ (e.g., the absence of the peaked-shape voltammograms characteristic of “battery-like” systems), and (ii) the GCD profiles were almost triangular (e.g., the absence of the voltage plateau verified for “battery-like” systems). In addition, the retention of capacitance was excellent even after 30,000 cycles (see Figure 7e,f). The small oscillations verified for the capacitance values can be ascribed to the progressive activation of the inner surface regions by deeper penetration of the electrolyte ions [36]. In any case, the coin cell maintained high capacitance retention (e.g., >95%) during the long-term charge–discharge experiments, which is typical for well-behaved supercapacitors. These important findings revealed that the solid-state redox process involving the nickel species in the hydrated oxide (e.g., gel layer structure) is highly reversible. 

A comparison of the VCs in Figure 7a,c,d revealed an enormous increase in the specific voltammetric current for the composite containing Ni(OH)_2_. For the applied scan rates of 50, 25, and 5 mV s**^−^**^1^, the specific capacitances obtained for CNF electrodes were 4.2, 5.0, and 5.6 F g**^−^**^1^, respectively. At the same time, the corresponding values for the NiO@CNF electrodes were 500, 520, and 620 F g**^−^**^1^, respectively. These findings are similar to those for Ni-based oxide electrode materials used in SCs [1,8,9,10,11,12]. As previously verified by Fantini and Gorenstein [57], the CV profiles for nickel hydroxide thin films are practically identical in neutral and alkaline solutions. Therefore, the absence in the present study of the pronounced peaks/bands commonly verified for battery-like Ni(OH)_2_ composite materials in alkaline solutions can be understood considering the occurrence of a strong synergism in the composite, i.e., the pseudocapacitive process was spread over the entire voltage window where the electrolyte is stable due to the presence of a uniform distribution in the porous electrode structure of reactive sites with different activation energies for the charge transfer reaction [3,28,33]. As a result, the electrochemical response is quite similar to that commonly verified for EDLCs. These are very important findings since, in most studies, the authors are interested in fabricating a composite material that exhibits the intrinsic characteristics of well-behaved supercapacitors. This featureless voltammetric profile verified for a hydrated transition metal oxide (TMO) is not new. For example, similar findings are commonly verified for the well-known RuO_2_⋅xH_2_O system in acidic and neutral solutions [58]. In addition, in the case of two-electrode cells, the faradaic current is distributed on a more flattened pattern in the voltammetric curves since the voltage for the working electrode (anode) is not measured against a true reference electrode having a potential benchmark that is constant during the experiments.

The analysis of Figure 7b evidenced a huge increase in the discharge time when Ni(OH)_2_ was incorporated in the carbon-based scaffold. As a result, the specific capacitance was strongly increased. Table 4 gathers the specific capacitances determined using the GCD findings for the different electrodes and normalizing factors. As seen, the overall specific capacitance for the two-electrode (coin cell) case is in good agreement with that verified in the EIS study (see Table 2). We chose to use the specific capacitance values corresponding to the individual (single) electrodes (e.g., the specific capacitances obtained in three-electrode cells) in addition to the two-electrode case obtained for the symmetric coin cell. 

Obviously, the specific pseudocapacitance values of practical importance are those referring to the two-electrode case, as in the current EIS study. However, to facilitate a comparison with the literature regarding the use of traditional three-electrode cells, we calculated the capacitances obtained from GCDs for the individual electrodes. In this sense, using the experimental masses of 12.3 mg (CNFs) and 12.6 mg (composite), we verified a very small specific capacitance of 16 F g^−^^1^ for CNFs. On the contrary, we verified an enormous specific pseudocapacitance of 2548 F g^−^^1^ for NiO/Ni(OH)_2_ that is very close to the predicted theoretical value of ~2600 F g^−^^1^ [59]. The maximum specific energy and power obtained for the NiO@CNF composite electrodes were 88.47 Wh kg^−^^1^ and 20,833 W kg^−^^1^, respectively. The maximum specific energy and power obtained for the CNF electrodes were 0.56 Wh kg^−^^1^ and 96.78 W kg^−^^1^, respectively. These GCD findings were obtained at 1 mA.

It is worth mentioning that it is not correct to normalize the capacitance using very small electrode masses for extrapolation purposes since the electrochemical data probably would not scale up with the larger masses used in practical electrodes [60]. Moreover, we would like to emphasize that, in several studies, the pseudocapacitance values reported for single- and two-electrode cases are illusory (incorrect), i.e., different battery-like electrochemical systems were erroneously characterized as true pseudocapacitors. As recently discussed by some of the present authors [1], this important issue arises from the incorrect distinction between the *capacity* (Ah g^−^^1^) and *capacitance* (F g^−^^1^) concepts [61].

## 4. Conclusions

Different symmetric supercapacitors (EDLC and PC) were used as “model systems” to apply a robust impedance model to understand the influence of disordered electrode materials on the transport anomalies occurring in the ionic and electronic conductors. For the first time, deviations from Fick’s law were identified and quantified during the charge storage process in SCs using the electrochemical impedance spectroscopy (EIS) technique. The use of an impedance model containing two time constants was adequate to represent normal and anomalous charge transports in the different channels (phases) in intimate contact. 

Abnormal charge transport in the ionic and electronic conductors was quantified by the dispersive parameters (*n* and *s*) extracted from the different time constants. The anomaly degree verified for the ionic transport inside the narrow pores was pronounced for the EDLC system (*n* = 0.42). In contrast, in the case of the NiO@CNF (composite) system, the ionic transport was practically regular (*n* = 0.92). At the same time, the electronic transport was nearly regular (*s* ≥ 0.98) for the different solid-state conductors. The analysis of the exponent (*β*) representing the capacitance and pseudocapacitance dispersions revealed a low degree of deviation (*β* ≈ 0.95) for the different electrode materials compared to the ideal case (*β* = 1). 

It was verified that the specific capacitance increased from 2.62 to 536 F g^−1^ after decorating the carbon substrate (CNF—carbon nanofibers) using NiO nanoparticles. These findings support the occurrence of a strong synergism in the composite, where the porous electrode structure of CNFs propitiates a fast ionic transport towards the hydrated oxide structure (NiO⋅*x*H_2_O) where the solid-state pseudocapacitance is localized. In addition, CNFs act as localized current collectors that promote a rapid capture of the electrons that originated from the solid-state redox reaction. Voltammetric and galvanostatic studies corroborated the very good capacitance and pseudocapacitance behaviors exhibited by the different symmetric coin cells, i.e., the presence of rectangular voltammetric profiles in conjunction with the triangular galvanostatic charge–discharge curves confirmed the strong capacitive behavior observed in the EIS study. Obviously, charge transport anomalies can only be identified using the impedance technique.

## Figures and Tables

**Figure 1 nanomaterials-12-00676-f001:**
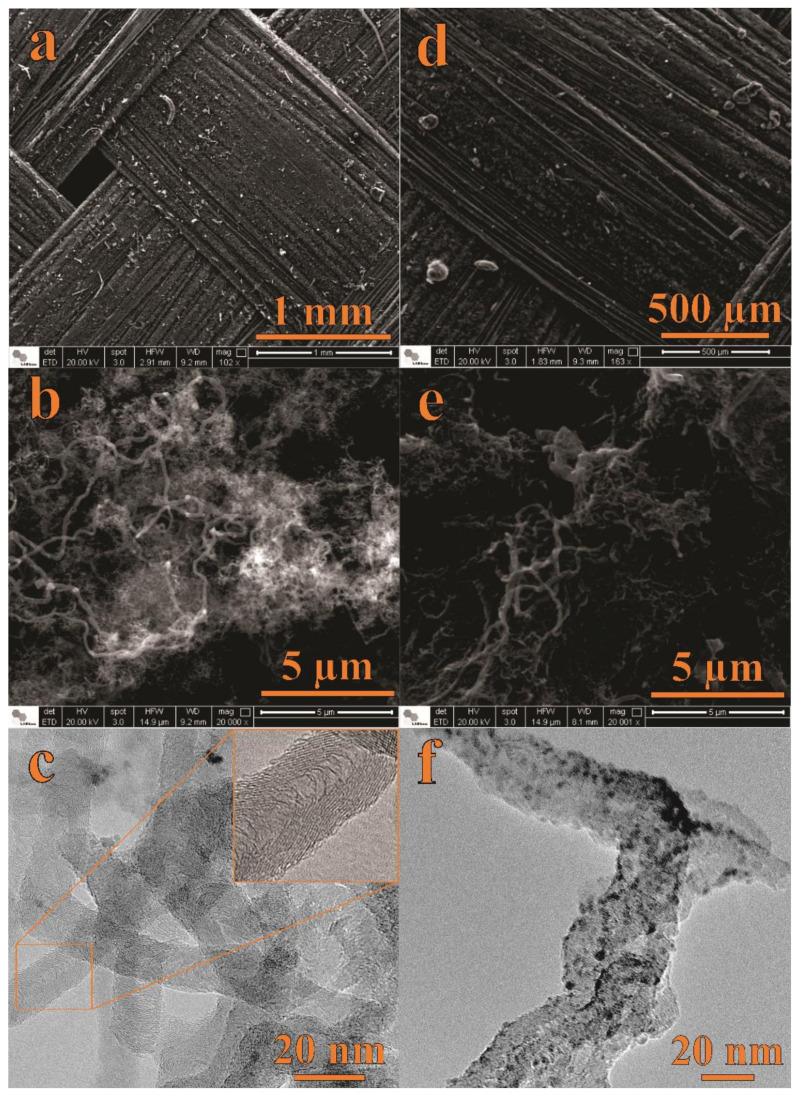
SEM and TEM data of CNFs (**a**–**c**) and NiO@CNFs (**d–f**).

**Figure 2 nanomaterials-12-00676-f002:**
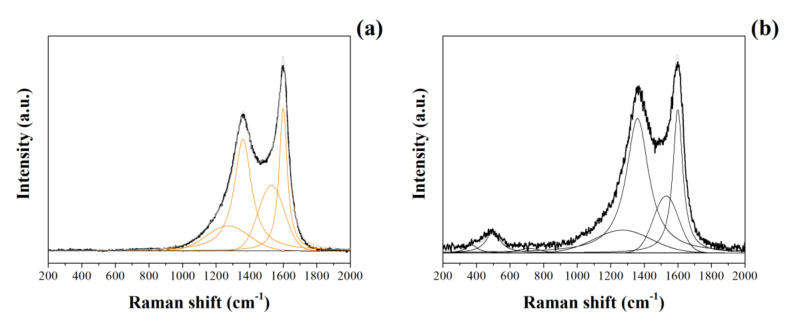
Raman spectra of CNFs (**a**) and NiO@CNFs (**b**) obtained at 488 nm.

**Figure 3 nanomaterials-12-00676-f003:**
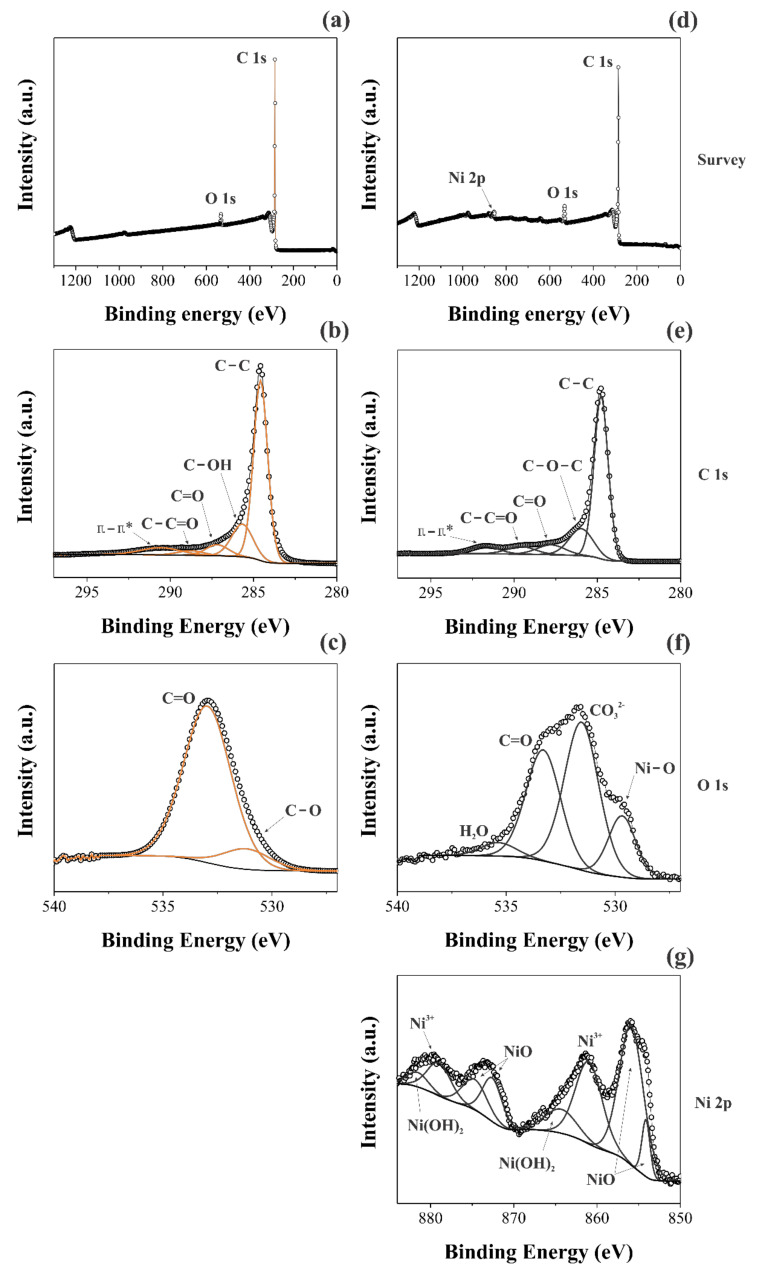
XPS spectra took for the CNFs (**a–c**) and NiO@CNFs (**d–g**) samples.

**Figure 4 nanomaterials-12-00676-f004:**
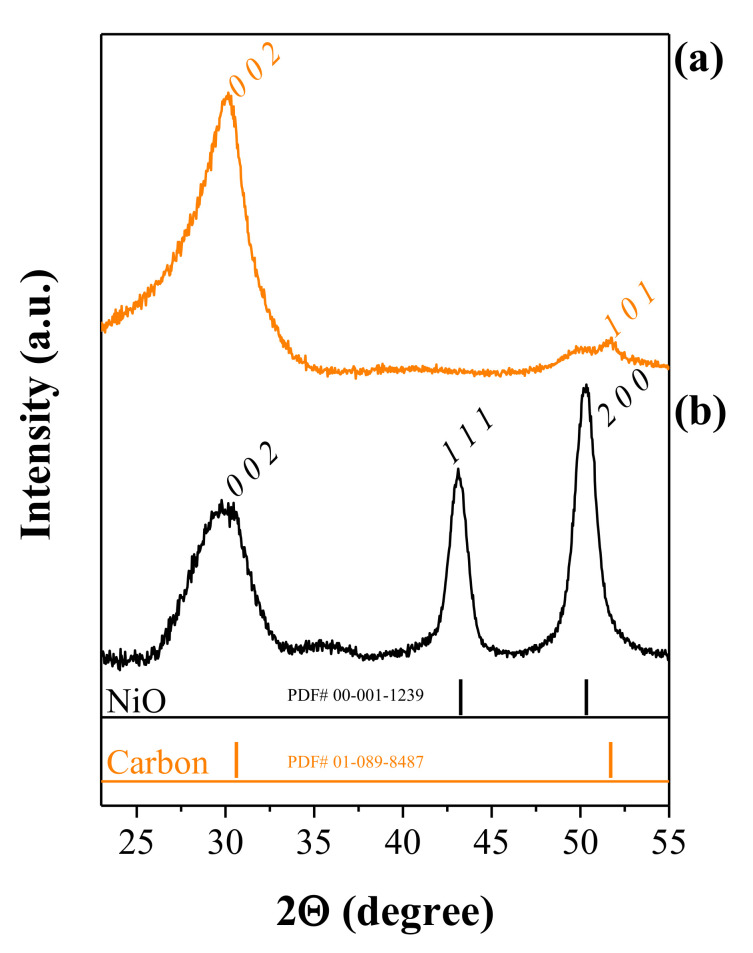
XRD patterns of (**a**) CNFs and (**b**) NiO@CNFs.

**Figure 5 nanomaterials-12-00676-f005:**
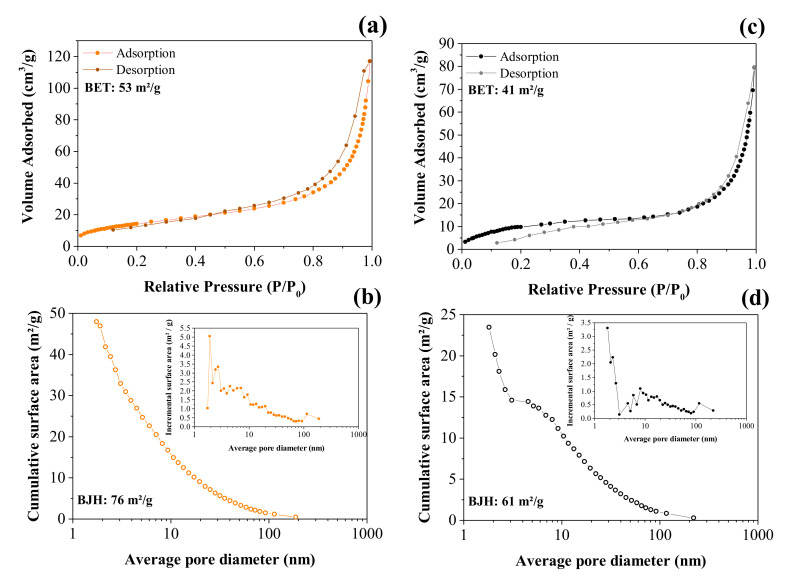
BET and BJH analyses of the adsorption/desorption nitrogen isotherm curves obtained for the CNFs (**a**,**b**) and NiO@CNFs (**c**,**d**).

**Figure 6 nanomaterials-12-00676-f006:**
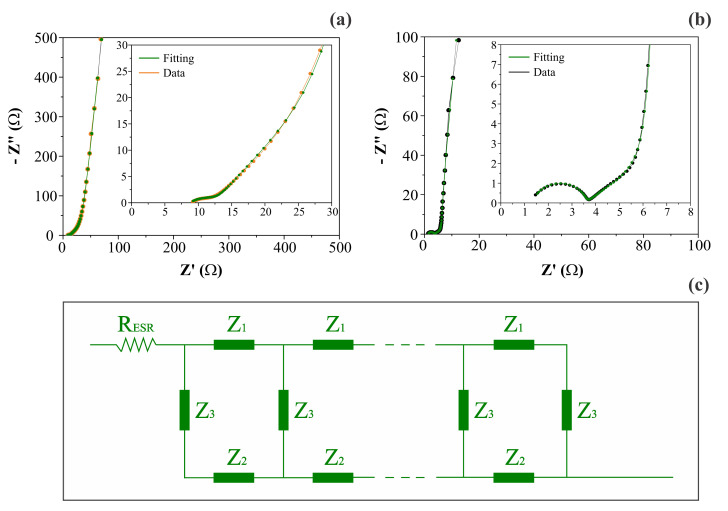
Complex plane plots (**a**,**b**) and the generic double-channel transmission line model (**c**), including the anomalous transport for porous/disordered electrodes. Plots (**a**) and (**b**) refer to CNF and NiO@CNF electrodes, respectively. Impedances *Z*_1_ and *Z*_2_ are composed of a circuit containing an ohmic resistor (*R*) parallel to a constant phase element (CPE). A CPE represents the interface (*Z*_3_) impedance.

**Figure 7 nanomaterials-12-00676-f007:**
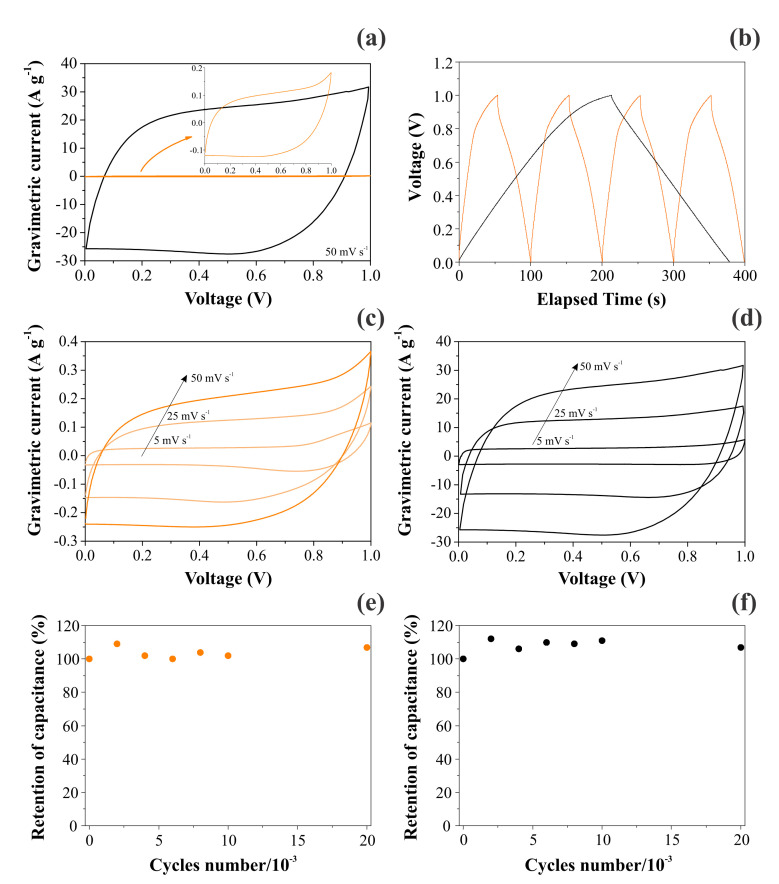
Electrochemical findings for CNFs (left) and NiO@CNFs (right) electrodes in symmetric coin cells filled with a 1.0 M Li_2_SO_4_ aqueous solution: (**a**,**c**,**d**) the scan rate is indicated in the figure; (**b**) GCD profiles obtained at 0.4 A g^−1^ for CNFs and 4 A g^−1^ for NiO@CNFs; (**e**,**f**) the cyclability test performed at 20 A g^−1^ for both cells. All experiments were carefully performed using the capacitive cell potential/voltage range where the electrolyte was stable (e.g., in the absence of water electrolysis).

**Table 1 nanomaterials-12-00676-t001:** XPS data obtained for the CNFs and NiO@CNFs samples.

Sample	Element	Functional Groups	B.E./eV
CNFs	C1s	C-C	284.6
		C-OH	285.6
		C=O	287.2
		O-C=O	288.8
		*π–π**	290.6
	O1s	C-O	531.3
		C=O	533.0
NiO@CNFs	Ni2p	NiO	854.2
		NiO	856.0
		Ni^3+^	861.1
		NiO(OH)_2_	864.6
		NiO	872.8
		NiO	875.0
		Ni^3+^	879.2
		NiO(OH)_2_	881.8
	C1s	C-C	284.8
		C-O-C	286.0
		C=O	287.8
		C-C=O	289.6
		*π–π**	291.8
	O1s	Ni-O	529.7
		CO_3_^2−^	531.6
		C=O	533.3
		H_2_O	535.4

**Table 2 nanomaterials-12-00676-t002:** Impedance parameters obtained from the CNLS simulation according to the effective macro homogeneous description of two closely mixed phases.

Electrodes	*R*_ESR_ (Ω g)	*R*_ionic_ (Ω g^−1^)	*Q*_ionic_ (F s*^n^*^−1^ g)	*R*_electronic_ (Ω g^−1^)	*Q*_electronic_ (F s*^s^*^−1^ g)	*Q*_edl_ or *Q*_pc_ * (F s*^β^*^−1^ g^−1^)
CNFs	0.21	3.48 × 10^3^	1.45 × 10^−4^ (*n* = 0.42)	2.16 × 10^2^	4.92 × 10^−4^ (*s* = 1.00)	2.62 (*β* = 0.95)
NiO@CNFs	0.04	4.32 × 10^2^	2.22 × 10^−7^ (*n* = 0.92)	1.11 × 10^2^	3.02 × 10^−3^ (*s* = 0.98)	5.36 × 10^2^ (*β* = 0.96)

* *Q*_pc_ is the pseudocapacitance for NiO@CNFs, while *Q*_edl_ is the electrical double-layer capacitance for CNFs.

**Table 3 nanomaterials-12-00676-t003:** Characteristic frequencies and time constants for the ionic (*Z*_1_) and electronic (*Z*_2_) transports according to the theoretical impedance model represented by Equation (1). The exponents *n* and *s* are obtained from Equations (3) and (4), respectively.

Electrode	*f*_c_(*Z*_1_)/s^−1^	*f*_c_(*Z*_2_)/s^−1^	*τ*_c_(*Z*_1_)/s	*τ*_c_(*Z*_2_)/s
CNFs	0.81 (*n* = 0.42)	1.50 (*s* = 1.00)	1.25 (*n* = 0.42)	0.67 (*s* = 1.00)
NiO@CNFs	3.71 × 10^3^ (*n* = 0.92)	0.49 (*s* = 0.98)	2.70 × 10^−4^ (*n* = 0.92)	2.06 (*s* = 0.98)

**Table 4 nanomaterials-12-00676-t004:** Overall specific capacitances in the unities of F g^−1^ obtained from the GCD findings.

Conditions	NiO@ CNFs	CNFs
Single electrode	2548	16
Two electrodes	637	4

## Data Availability

The data is available on reasonable request from the corresponding author.

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
