# Peer review of "Electrochemical Behavior of Symmetric Electrical Double-Layer Capacitors and Pseudocapacitors and Identification of Transport Anomalies in the Interconnected Ionic and Electronic Phases Using the Impedance Technique"

_nanomaterials, 2022, doi:10.3390/nano12040676_

Round 1

Reviewer 1 Report

Research in Supercapacitors is evolving these days as it is a significant energy storage device. Various electrode materials including metal oxides, nitrides, carbon, and conducting polymers have been developed for supercapacitor applications. Poor electrical conductivity and mechanical instability are two major obstacles to realizing the high performance of either TMOs or carbonaceous-based materials. Therefore, the authors have demonstrated the mechanism involved in decorated carbon-based TMOs (NiO@CNFs) exhibiting both EDLCs and battery kind of redox mechanisms. This is a reasonable contribution with good materials engineering and impedance insights; however, some parts of the manuscript (text and discussion) must be improved. All the formulas and units are dreadful. A major revision is required before rendering a final decision.

My specific comments are below:

  • The charge storage mechanism of NiO@CNFs is detailed recently by authors in Nanoscale 10.1039/D1NR00065A; being this the case what is the novelty of this work?
  • Introduction first few lines (39 – 46) what are those instructions, makes no sense.
  • Electrochemical technique (EIS) Nyquist plot of impedance spectra in lines 71 – 79: Some electrochemical insights on the Nyquist plot (using the background cited in the literature 10.1039/C8NR03824D) will assist the readers to understand the impedance and its role on the mass transport aspects and ohmic resistances; how it influences the capacitive behaviour.
  • There are lots of typos throughout the manuscript; very ugly. Suffix and prefix (in the units and formula) need to be strictly maintained.
  • Few of the sections in the introduction can be trimmed, too lengthy.
  • The relevant work reported in Ni oxide (XPS) peaks and their electrochemical behaviour for the supercapacitor studies reported in the literature (doi.org/10.1021/acsami.1c16287;) can be discussed to ensure the charge-transport anomalies during the charge-storage process
  • How is the reported specific capacitance calculated?
  • In Fig. 7f, double-check the cycle numbers and the units? Is it 10-3?
  • What is the role of cation and anion (Li and S) mechanism in Li2SO4 electrolyte?
  • Line 837 – 843 long sentences, the intended meaning is unclear.
  • Why the values for CNF are as low as 4, 5, and 6 F/g?

Author Response

Comments and Suggestions for Authors

Research in Supercapacitors is evolving these days as it is a significant energy storage device. Various electrode materials including metal oxides, nitrides, carbon, and conducting polymers have been developed for supercapacitor applications. Poor electrical conductivity and mechanical instability are two major obstacles to realizing the high performance of either TMOs or carbonaceous-based materials. Therefore, the authors have demonstrated the mechanism involved in decorated carbon-based TMOs (NiO@CNFs) exhibiting both EDLCs and battery kind of redox mechanisms. This is a reasonable contribution with good materials engineering and impedance insights; however, some parts of the manuscript (text and discussion) must be improved. All the formulas and units are dreadful. A major revision is required before rendering a final decision.

General authors’ response: 

Thank you for your comments. We improved all manuscript parts necessary to attend to the Reviewer’s comments. We want to emphasize that our intention with this article is to present a real contribution to the field of Supercapacitors to stimulate a rational impedance analysis based on Porous Electrode Models. We avoided the commonly found speculative theoretical discussion of irrational (illusory) findings perpetuated in the literature by different authors.

Regarding the formulas and unities commented by the Reviewer, several inconsistencies regarding unities, exponents, suffixes, prefixes, etc., resulted from the Journal’s Template model used by us, i.e., the insertion of the original Word file in the template resulted in severe formation losses. Therefore, the Revised Manuscript was sent to the journal using the original formation of the Word document.

P.S. The presented formulas and unities in their correct format (Revised Manuscript) agree with the style used by the specialist in the EIS field. Generally, we followed in this work the most advanced impedance models and the formulation therein, as are the seminal works by Bisquert et al., Lasia et al., etc. 

P.S. A native English speaker revised the text to avoid typos and spelling errors.

My specific comments are below:

  • The charge storage mechanism of NiO@CNFs is detailed recently by authors in Nanoscale 10.1039/D1NR00065A; being this the case what is the novelty of this work?

Authors’ response:

The mentioned article does not contain any EIS study. It is a complementary source of information solely based on Raman and DFT analyses. Therefore, the novelty of the current work is quite evident since it is based on a comprehensive EIS study.

  • Introduction first few lines (39 – 46) what are those instructions, makes no sense.

Authors’ response:

We apologize for this technical error due to compilation issues using the Journal’s Template during the preparation of the manuscript for submission. The inappropriate sentences were eliminated.

  • Electrochemical technique (EIS) Nyquist plot of impedance spectra in lines 71 – 79: Some electrochemical insights on the Nyquist plot (using the background cited in the literature 10.1039/C8NR03824D) will assist the readers to understand the impedance and its role on the mass transport aspects and ohmic resistances; how it influences the capacitive behaviour.

Authors’ response:

Thank you for indicating this article. We included it in the Introduction section.

  • There are lots of typos throughout the manuscript; very ugly. Suffix and prefix (in the units and formula) need to be strictly maintained.

Authors’ response: 

Thank you for the pertinent comments. We corrected typos, suffixes, etc., as recommended. As already mentioned, several technical errors appeared due to compilation issues using the Journal’s Template.

  • Few of the sections in the introduction can be trimmed, too lengthy.

Authors’ response: 

As suggested, we reduced the Introduction.

  • The relevant work reported in Ni oxide (XPS) peaks and their electrochemical behaviour for the supercapacitor studies reported in the literature (doi.org/10.1021/acsami.1c16287;) can be discussed to ensure the charge-transport anomalies during the charge-storage process

Authors’ response:

Thank you for the pertinent suggestion. The following sentences extracted from the suggested reference were included in the current manuscript. “Sharma et al. (doi.org/10.1021/acsami.1c16287) recently reported on tuning the nanoparticle interfacial properties and stability of the core-shell structure in Zn-doped NiMoO4@AWO4 electrodes. It was considered Zn-doped nickel molybdate (NiMoO4) (ZNM) as a core crystal structure and AWO4 (A = Co or Mg) as a shell surface. These authors verified the ability to tune the core-shell nanocomposites with a surface reconstruction as a source for surface energy (de)stabilization. It was verified the performance of the core@shell is significantly affected by the chosen intrinsic properties of metal oxides with high performance compared to a single-component system in supercapacitors. The constructed asymmetric device (e.g., Zn-doped NiMoO4@CoWO4 (ZNM@CW)||activated carbon) exhibited a superior pseudo-capacitance, delivering a high areal capacitance of 892 mF cm-2 at 2 mA cm-2 and excellent cycling stability (e.g., 96% capacitance retention after 1000 charge-discharge cycles). In this sense, the study conducted by Sharma et al. (doi.org/10.1021/acsami.1c16287) presented different theoretical and experimental insights with the extent of the surface reconstruction to explain the storage properties in SCs.”

  • How is the reported specific capacitance calculated?

Authors’ response:

The text was slightly improved. In any case, this information was already available in the submitted manuscript, lines 274-279. As can be verified:  “The overall masses of the CNF (model EDLC system) and NiO@CNF (model PC system) electrodes housed in the different symmetric coin cells were 12.3 and 12.6 mg, respectively….The different intensive parameters (gravimetric quantities) reported in the electrochemical study were calculated using these masses.”

  • In Fig. 7f, double-check the cycle numbers and the units? Is it 10-3?

Authors’ response:

Thank you for verifying this error. The correct unit is obtained when “Cyclic Number/10+3” instead of “Cyclic Number/10-3”.

  • What is the role of cation and anion (Li and S) mechanism in Li2SO4 electrolyte?

Authors' response:

As discussed in lines 596-617, the Li+-ions can participate in a 'substitutive solid-state redox' (parasitic) reaction represented by Eq. (8) occurring parallel to the main intercalation/deintercalation process involving protons described by Eq. (7). On the contrary, the HSO4- and SO42- anionic species are present in the electrolyte as "spectator ions" during the pseudocapacitive charge-storage process. Obviously, in the electrical double-layer (electrostatic) charge-storage process, always present in any supercapacitor, these anionic species are responsible for the negative surface excess (Gibbs's Gamma Function) formed during the cathodic polarization process. At the same time, the Li+-ions are accountable for the positive surface excess formed during anodic polarization.

  • Line 837 – 843 long sentences, the intended meaning is unclear.

Authors' response:

Thank you for the comment. These unnecessary sentences were omitted in the revised manuscript.

  • Why the values for CNF are as low as 4, 5, and 6 F/g?

Authors' response:

As discussed in lines 439-446, BET findings revealed a low specific surface area of 53 m2/g for CNFs with a significant contribution of mesopores. In comparison, good-quality AC electrodes with ca. 2000 m2/g commonly yield specific electrostatic capacitances of ca. 150-200 F/g. As a result, the active surface area is very low in the current study, resulting in reduced capacitances in the 4-6 F/g range. Therefore, as already explained, the contribution of the surface ionic accumulation/depletion at the interface (e.g., electrostatic contribution) to the overall specific capacitance (electrostatic + pseudocapacitance) is very limited, i.e., the pseudocapacitive process due to the solid-state redox reactions (see Eq. (7) and Eq. (8)) dominates the overall charge-storage process in the composite electrode.

Reviewer 2 Report

The manuscript describes the method to prepare an “electrochemical behavior of symmetric electrical double-layer capacitors and pseudocapacitors and identification of transport anomalies in the interconnected ionic and electronic phases using the impedance technique”. The structural, morphological, and electrochemical properties were studied, along with this they tested supercapacitor properties. There are a number of issues in the current version of the manuscript that I have mentioned in the following detailed comments. Hence it cannot be accepted for publication in its current version. I would suggest authors need to solve the following comments and re-submit again, I think this can be strongly accepted after resubmission.

  1. The abstract is too long, so the author should revise the abstract.
  2. Introduction part is too long, so the author should revise the Introduction, the introduction does not highlight its morphology.
  3. In XRD analysis, the curve is too smooth to show some peaks of the CNFs and NiO, and there is no standard card for comparison of the NiO and CNFs; Authors should provide pure NiO XRD. In addition, the Authors should add the standard card number of NiO.
  4. The author should add size of the nanostructure in SEM and Tem.
  5. Author should describe EIS, CV, CD to long, I think the author should revise all parts.
  6. The description of the electrochemical part does not match the standard data, the author should compare all results.
  7. All equations can be numbered uniformly
  8. Please avoid repetition.
  9. Conclusion should revised.

Author Response

Comments and Suggestions for Authors

The manuscript describes the method to prepare an “electrochemical behavior of symmetric electrical double-layer capacitors and pseudocapacitors and identification of transport anomalies in the interconnected ionic and electronic phases using the impedance technique”. The structural, morphological, and electrochemical properties were studied, along with this they tested supercapacitor properties. There are a number of issues in the current version of the manuscript that I have mentioned in the following detailed comments. Hence it cannot be accepted for publication in its current version. I would suggest authors need to solve the following comments and re-submit again, I think this can be strongly accepted after resubmission.

  1. The abstract is too long, so the author should revise the abstract.

Authors' response:

The Abstract was amended to improve its quality.

  1. Introduction part is too long, so the author should revise the Introduction, the introduction does not highlight its morphology.

Authors' response:

Thank you for the pertinent comment. As requested, the Introduction was strongly modified/reduced. We retained the main part regarding the use of the EIS technique and the best practices for studying supercapacitors, as well as the pitfall inherent to the misuse of the diffusive elements.

  1. In XRD analysis, the curve is too smooth to show some peaks of the CNFs and NiO, and there is no standard card for comparison of the NiO and CNFs; Authors should provide pure NiO XRD. In addition, the Authors should add the standard card number of NiO.

Authors' response: 

There is confusion arising in this point mentioned by the Reviewer. As shown in Fig. 4, the standard card for comparison was included for both NiO (PDF# 00-001-1239) and Carbon (PDF# 01-089-8487). As seen, the identification of the different phases can be determined unambiguously.

  1. The author should add size of the nanostructure in SEM and Tem.

Authors' response: 

There is confusion arising in this point mentioned by the Reviewer. Figure 1 includes the size bars for different images. Also, as shown in lines 353-357, one can read, “As verified from the TEM analysis, the modified CNFs are highly defective with a non-linear structure. The CNFs diameter range from 10 to 90 nm. At the same time, NiO exhibited quasi-spherical nanoparticles with diameters ranging from 1 to 5 nm.”

  1. Author should describe EIS, CV, CD to long, I think the author should revise all parts.

Authors' response:

The entire text was revised to remove eventual inconsistences. However, as clearly stated in the article’s title, the main focus of this work is based on the Electrochemical Studies, mainly the EIS part. The ex-situ characterization studies (SEM, TEM, Raman, XPS, XRD, and BET) were merely included in the present context for the readers to be confident with the different physicochemical properties exhibited by the EDLC (CNFs) and PC (NiO) materials used in the electrochemical experiments. In this sense, a reduction in length of the Electrochemical Section (EIS, CV, CD) is not beneficial considering the true content of the article. In fact, all relevant findings extracted for the electronic and ionic charge behavior during the charge-storage process, and the presence of charge transport anomalies, were obtained from the EIS findings.

  1. The description of the electrochemical part does not match the standard data, the author should compare all results.

Authors' response:

Electrochemical section was entirely revised in the search of conflicting points. We think that the revised version does not contain any conflicting findings.

  1. All equations can be numbered uniformly

Authors' response:

All equations were uniformly numbered.

  1. Please avoid repetition.

Authors' response:

The entire text was thoroughly revised to omit repetition.

  1. Conclusion should revised.

Authors' response:

The conclusion was revised and improved.

Round 2

Reviewer 1 Report

In this reviewer's opinion, the revised version is suitable to publish. The author's have fairly addressed the queries.